# How Can the Development of Digital Economy Empower Green Transformation and Upgrading of the Manufacturing Industry?—A Quasi-Natural Experiment Based on the National Big Data Comprehensive Pilot Zone in China

**Qiansheng Gong** [1,2,*], **Xiangyu Wang** [1,2] **and Xi Tang** [1,2]

1 School of Economics and Management, Xi'an Shiyou University, Xi'an 710065, China; wxy9811102022@163.com (X.W.); tx3281438@gmail.com (X.T.)
2 Shaanxi (University) Oil and Gas Resources Economics and Management Research Center, Xi'an Shiyou University, Xi'an 710065, China
\* Correspondence: gongqiansheng1979@126.com; Tel.: +86-136-0925-3723

**Abstract:** Using the panel data of the manufacturing industry in 30 provinces of China from 2005 to 2021, this research takes the establishment of a Chinese national-level comprehensive big data pilot zone as a quasi-natural experiment, empirically analyzes the processing effect of digital economy development on the green transformation and upgrading of manufacturing industry by using a time-varying DID model. The results show that the development of the digital economy can significantly promote the green transformation and upgrading of the manufacturing industry. Further analysis reveals that the development of the digital economy has a significant effect on the green transformation and upgrading of the manufacturing industry in regions with low economic development levels and regions with high network development levels. The development of the digital economy can significantly stimulate the green technology innovation of enterprises and promote the upgrading of industrial structures so as to promote the green transformation and upgrading of the manufacturing industry. Manufacturing agglomeration and environmental regulation intensity have moderating effects and threshold effects, respectively, on the impact of digital economy development on the green transformation and upgrading of the manufacturing industry.

**Keywords:** digital economy; manufacturing industry; green transformation and upgrading; national big data comprehensive pilot zone; time-varying DID model; total factor productivity

## 1. Introduction

As the Chinese economy shifts from the stage of high-speed growth to the stage of high-quality development, the traditional industrial structure and manufacturing system can no longer provide sustainable growth momentum for the economy, and it is urgent to find new economic growth points [1]. At the same time, faced with more and more resource consumption and increasingly severe environmental pollution problems, the optimization and upgrading of industrial structures have been imminent. As the "foundation of our country", the manufacturing industry has made outstanding contributions to the rapid growth of the Chinese economy for more than 40 years. According to statistics, the value added of China's manufacturing industry accounted for nearly 30% of GDP in 2021, ranking first in the world for many years. However, at the present stage, the Chinese manufacturing industry is still in the middle and low end of the global value chain division of the labor system, and extensive production mode causes great pressure on China's environmental protection [2]. Relevant statistics show that in 2019, energy consumption in China's manufacturing sector accounted for more than half of the country's total energy consumption, reaching a staggering 55%; carbon emissions accounted for 30% of the country's total carbon emissions, second only to the energy sector at 35%. It can be

seen that the manufacturing industry plays an absolutely dominant role in both energy consumption and carbon emissions. Therefore, under the background of the Chinese "3060" dual carbon strategy, the manufacturing industry urgently needs green transformation and upgrading. With the wide application of industrial Internet and big data technology, the rapid development of the digital economy can realize the optimization of development mode and technological innovation of the manufacturing industry [3], greatly promote the digitization process of the manufacturing industry, and facilitate the green transformation and upgrading and high-quality development of the manufacturing industry. At the same time, the digital development of the manufacturing industry has provided more jobs, increasing the demand for big data talents and providing opportunities for more job seekers.

With the rapid development of the digital economy, data resources, as a new factor of production, play an increasingly important role in international competition [4]. Countries around the world have formulated their own big data development strategies, hoping to enhance their comprehensive competitiveness by firmly controlling data resources. In this context, fully exploiting and releasing the dividends of data elements, vigorously developing the digital economy with big data as a distinctive feature, and empowering high-quality economic development have also become one of the important elements of China's economic development [5]. Therefore, in September 2015, China first proposed the establishment of the Guizhou Big Data Comprehensive Pilot Zone; in 2016, the pilot program was launched successively in Beijing, Tianjin, Hebei, Inner Mongolia, Shenyang, Shanghai, Guangdong, Chongqing, and other provinces, aiming to maximize the value of data resources, develop the emerging big data industry, and realize the rapid development of the digital economy. Based on this, this paper takes the establishment of a national-level big data comprehensive pilot zone as a quasi-natural experiment to study the processing effect of digital economy development on the green transformation and upgrading of the manufacturing industry.

The rest of this paper is arranged as follows: The second part is the literature review. The third part analyzes the theoretical mechanism and puts forward the research hypothesis. The fourth part is the research design, introducing the model, variables, and data. The fifth part gives the empirical results, including baseline regression results, heterogeneity analysis, mechanism analysis, etc. The sixth part summarizes the research conclusions and puts forward countermeasures and suggestions.

## 2. Literature Review

The academic community has paid great attention to the rapid development of the digital economy in recent years, and the existing literature on the digital economy has broadly experienced three stages: the information economy, the Internet economy, and the digital economy [6]. The development of the digital economy plays a significant role in promoting regional innovation [7], improving production efficiency [8], and optimizing enterprise performance [5]. Digital economy development helps to promote industrial structure upgrading [9,10] and achieve high-quality economic development [11]. With the rapid development of the digital economy, digital technology has been widely used in production, which has achieved partial labor substitution to some extent [12], thus improving production efficiency and achieving economic growth [13,14]. In a sense, digitalization has become an important way to transform and upgrade traditional industries. Therefore, how to use digitalization to achieve transformation and upgrading in the manufacturing industry, which is a traditionally advantageous industry in China, has gradually become an important topic for academic discussion. At present, the research findings in the transformation and upgrading of the manufacturing industry mainly focus on traditional research fields such as resource allocation [15], innovation investment [16], and industrial integration [17]. There are also some studies on green welfare [18] and the new development pattern of the "double cycle" [19], which take into account economic growth and environmental benefits. In recent years, some scholars have begun to conduct some

exploratory studies on the role of the digital economy in promoting the transformation and upgrading of the manufacturing industry. Some scholars discussed the driving role of the digital economy in the green development of the manufacturing industry from the perspectives of industry performance [20], geographical agglomeration [21], and global value chain [22]. Cai Yanze et al. (2021) and Yu Donghua and Wang Meijuan (2022), respectively, introduced factors such as innovation environment and entrepreneurship to analyze the mechanism of the digital economy on manufacturing upgrading [23,24]. Some other scholars have conducted in-depth discussions on the relationship between specific industries or enterprises [25], such as equipment manufacturing [26] and sports goods manufacturing [27], and the digital economy, deepening the logical connection between the digital economy and the manufacturing industry. In addition, in the existing literature, it is common to take the digital economy [1,4], environmental regulation [28], and other policies issued by the government as quasi-natural experiments and to carry out research by applying the DID model.

In summary, the existing literature provides useful ideas for subsequent research. However, the research on the impact of the digital economy on the transformation and upgrading of the manufacturing industry mainly stays at the economic level, and fewer environmental factors were included to further discuss how to realize the green transformation and upgrading of the manufacturing industry through the development of the digital economy. Therefore, this paper plans to use panel data of the manufacturing industry in 30 provinces of China from 2005 to 2021, taking into account both economic growth and environmental performance, and innovatively introduce the establishment of pilot zones as a quasi-natural experiment into the empirical analysis of the impact of digital economy development on green transformation and upgrading of manufacturing industries based on the promotion of digital economy development by the construction of Chinese national-level big data comprehensive pilot zones [29] to provide a realistic basis for the manufacturing industry to achieve a higher level of development.

## 3. Theoretical Mechanism and Research Hypothesis

This research analyzes the impact mechanism of digital economy development on the green transformation and upgrading of the manufacturing industry from two dimensions, including direct effect and indirect effect, and proposes research hypotheses accordingly.

### 3.1. Direct Effect

With the rapid development of the industrial Internet and big data technology, the digital economy has gradually become the new driving force of Chinese industrial transformation. The manufacturing industry, as a traditional superior industry in China, has also embarked on the road of digital development. On the one hand, the digital economy with data elements as the core has positive externalities in promoting the integration and development of the emerging big data industry and traditional manufacturing industry and optimizing resource allocation [24]; on the other hand, under the background of green and sustainable development advocated by the state, the direction of transformation and upgrading of the manufacturing industry is to enhance the proportion of "green genes", so that the green development-oriented manufacturing industry continue to emerge [30]. With its advantages of networking and data, the digital economy can provide technical support for the green transformation of manufacturing production mode and realize the green transformation and upgrading of the manufacturing industry. Based on the above analysis, the research Hypothesis 1 is proposed:

**Hypothesis 1.** *The development of the digital economy can significantly promote the green transformation and upgrading of the manufacturing industry.*

*3.2. Indirect Effects*

(1) Technological Innovation Effect

With the popularization of the Internet, the asymmetry in information acquisition is becoming less and less, which is more conducive to the integrated development, innovation, and upgrading of the industry to a certain extent. In the context of the rapid development of big data technology, the agglomeration of production factors such as capital, talents, and technology has significantly improved the level of urban technological innovation [31]. According to the new growth theory, knowledge is the source of technological innovation. The development of the digital economy broadens the transmission channels and ways of knowledge, realizes the digital development of production technology, and thus provides great convenience for enterprises' technological innovation. Meanwhile, given the important role of technological innovation in reducing pollution [32], the key to achieving the green transformation and upgrading of the manufacturing industry lies in how to use knowledge for green-oriented production technology innovation. Based on the above analysis, the following research Hypothesis 2a is proposed:

**Hypothesis 2a.** *The development of the digital economy can achieve the green transformation and upgrading of the manufacturing industry by inducing green technological innovation.*

(2) Industrial Structure Effect

As a new driving force for industrial transformation and upgrading, the widespread application of digital technology promotes the rapid development of emerging industries. It promotes the rise of new products and business models through the integration of the industrial chain and innovation chain, thus greatly enhancing the correlation between industries and finally forming the technology diffusion effect. With the development of the digital economy, the traditional crude production mode can also be reborn through the organic integration of new and traditional industries [33], promoting the upgrading of industrial structure and greatly increasing the proportion of clean, efficient, and intelligent production modes in the industry. Therefore, the upgrading of industrial structure is consistent with the green transformation and upgrading of the manufacturing industry in concept, and industrial structure optimization and upgrading can also significantly promote the green transformation and upgrading of the manufacturing industry. Based on the above analysis, the research Hypothesis 2b is proposed:

**Hypothesis 2b.** *The development of the digital economy can realize the green transformation and upgrading of the manufacturing industry by promoting the upgrading of industrial structures.*

(3) Industrial Agglomeration Effect

With the continuous development of the manufacturing industry, the agglomeration of manufacturing enterprises and their related production factors in a specific region is continuously strengthened, resulting in positive externality, so as to promote the transformation and upgrading of the manufacturing industry through the division of labor and cooperation among enterprises [34]. Empowered by the digital economy, through the organic integration of industrial and innovation chains, advantages such as clear division of labor, cost reduction, and information sharing can be gradually formed, and positive external benefits can be generated. On the contrary, in the absence of scientific and reasonable planning and coordination, the enhancement of agglomeration may lead to "diseconomies of scale", resulting in vicious competition and management disorder, thus aggravating resource loss and environmental pollution, which is not conducive to the effective performance of the development of digital economy in promoting the green transformation and upgrading of the manufacturing industry. Based on the above analysis, research Hypothesis 3 is proposed:

**Hypothesis 3.** *Manufacturing agglomeration has a moderating effect between the development of the digital economy and the green transformation and upgrading of the manufacturing industry.*

(4) Environmental Regulation Effect

By combing the existing literature, it is found that according to the "cost-following effect", when the intensity of environmental regulation is low, the production cost of products will be increased, the profits of enterprises will be reduced, and the price of products will be raised, which will have a crowding out effect on green technology and clean equipment, resulting in high input cost and insufficient available funds for the development of the digital economy. The development of the digital economy has no significant promoting effect on the green transformation and upgrading of the manufacturing industry. On the contrary, when the intensity of environmental regulations is high, on the one hand, according to the "pollution refuge hypothesis", polluting enterprises in areas with strict regulations often choose to move to areas with relatively loose regulations in order to reduce treatment costs, so as to realize the transformation and upgrading of the manufacturing industry in areas with strict regulations. On the other hand, according to Porter's hypothesis, environmental regulation has an innovation compensation effect and can stimulate enterprises to carry out green-oriented technological innovation [35], thus promoting the green transformation and upgrading of the manufacturing industry. Based on the above analysis, the following research Hypothesis 4 is proposed:

**Hypothesis 4.** *Environmental regulation intensity has a threshold effect between the development of the digital economy and the green transformation and upgrading of the manufacturing industry.*

The theoretical mechanism analysis framework of this research is shown in Figure 1:

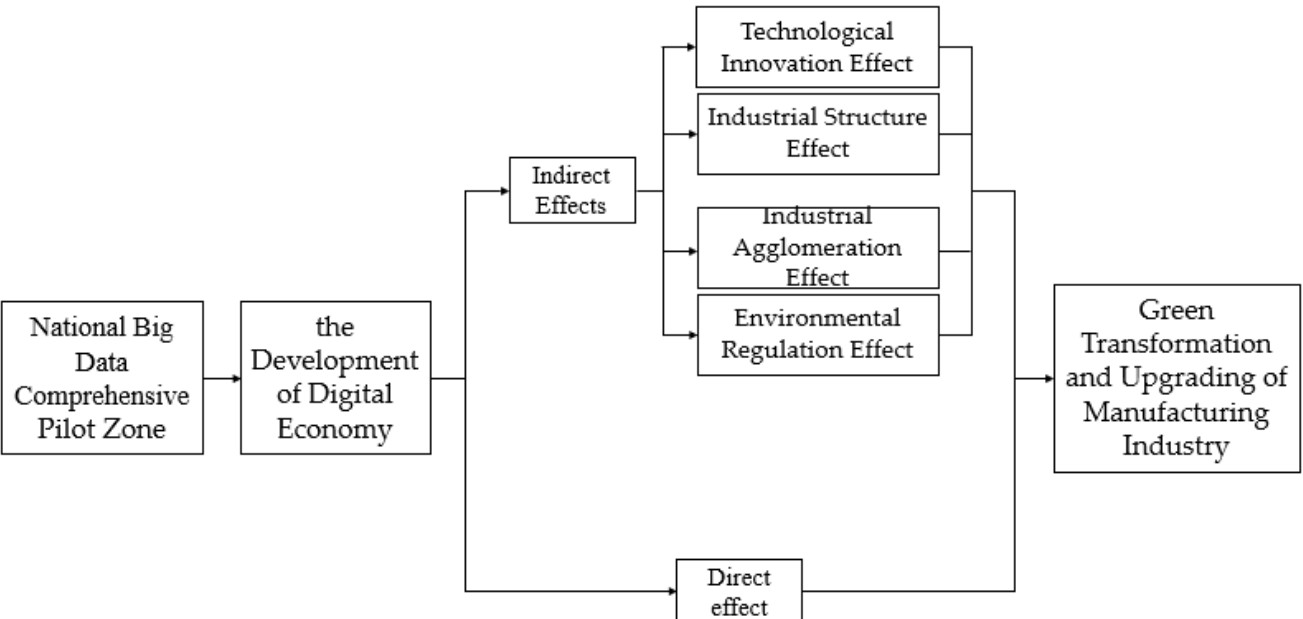

**Figure 1.** Theoretical mechanism analysis framework.

## 4. Study Design

### 4.1. Model Construction

The DID model is an important method to evaluate the policy effect; its independent variable is generally a grouping dummy variable. Specifically, the individuals affected by the policy are called the treatment group, and its grouping dummy variable will take 1. The individuals not affected by the policy are called the control group, and the grouping dummy variable will be 0. By comparing the different influences between the treatment

group and the control group before and after the introduction of the policy, it can eliminate the factors that do not change with time and cannot be observed and strip the policy treatment effect from it so as to achieve the purpose of policy evaluation.

In order to test whether the development of the digital economy can promote the green transformation and upgrading of the manufacturing industry, this paper refers to the method of Qiu Zixun (2021) [1] and takes the establishment of national-level big data comprehensive pilot zones as a quasi-natural experiment, with 30 provinces in China as the sample, and six provinces (Guizhou, Hebei, Inner Mongolia, Liaoning, Henan, and Guangdong) that carried out pilot zones and four municipalities (Beijing, Tianjin, Shanghai, and Chongqing) that carried out pilot zones as the treatment group while the other provinces as the control group. In terms of the time node selection of pilot areas, since the construction of the pilot zone in Guizhou Province started in 2015, and other provinces were only approved to carry out in 2016, this paper set the year of policy implementation in Guizhou Province as 2015 and the year of policy implementation in other pilot provinces as 2016 based on the practice of Guo Bingnan (2022) [4].

Since the year of the policy implementation in the pilot provinces is not a single year, it is difficult for the conventional DID model to assess the treatment effect of the pilot zone pilot. For this reason, this research refers to the method of Back T et al. (2001) to verify the impact of the digital economy on the green transformation and upgrading of the manufacturing industry by constructing a time-varying DID model [36]. The specific model is set as follows:

$$mgtfp_{it} = \alpha_0 + \alpha_1 D_{it} + \alpha_2 X_{it} + \mu_i + \delta_t + \varepsilon_{it} \tag{1}$$

where $mgtfp_{it}$ denotes the level of the green transformation and upgrading of the manufacturing industry in province $i$ in year $t$, which is measured by using green total factor productivity of the manufacturing industry in this paper; $D_{it}$ denotes the policy dummy variable established by the big data comprehensive pilot zone, and its coefficient $\alpha_1$ denotes the treatment effect of the policy; $X_{it}$ denotes the control variable; $\mu_i$ denotes the province fixed effect; $\delta_t$ denotes the year fixed effect; $\varepsilon_{it}$ denotes the random disturbance term.

### 4.2. Data Description

To ensure the availability of data and the consistency of statistical caliber, this paper studies the impact of digital economy development on the green transformation and upgrading of China's manufacturing industry based on the panel data of the manufacturing industry in 30 provinces of China from 2007 to 2021. The main data sources of this paper include China Statistical Yearbook, Chinese Provinces Statistical Yearbooks, China Environmental Statistical Yearbook, China Energy Statistical Yearbook, EPS Database, and WIND Database.

### 4.3. Variable Setting

(1) Explained Variable

Green total factor green productivity of manufacturing industry ($mgtfp$). According to theoretical mechanism analysis, this research adopts green total factor productivity, which covers production factors such as manpower, capital, and technology, and considers economic and environmental performance so as to fully reflect the green transformation and upgrading. Based on the ideas and practices of Li Xiaoyang et al. (2022) and Yang Xiang et al. (2019), this research calculates the green total factor productivity of the manufacturing industry by using the SMB-GML index method based on undesired output and comprehensively reflects the green transformation and upgrading of the manufactur-

ing industry from both environmental and economic aspects [2,37]. The GML index is constructed as follows:

$$GML_t^{t+1} = \frac{1+\underset{s_V^G}{\rightarrow}\left(x^t,y^t,b^t;g^t\right)}{1+\underset{s_V^G}{\rightarrow}\left(x^{t+1},y^{t+1},b^{t+1};g^{t+1}\right)} = GEFFCH_t^{t+1} \times GTECH_t^{t+1}$$

$$= \frac{1+\underset{s_V^t}{\rightarrow}\left(x^t,y^t,b^t;g^t\right)}{1+\underset{s_V^{t+1}}{\rightarrow}\left(x^{t+1},y^{t+1},b^{t+1};g^{t+1}\right)} \times \frac{1+\underset{s_V^G}{\rightarrow}\left(x^t,y^t,b^t;g^t\right)/1+\underset{s_V^t}{\rightarrow}\left(x^t,y^t,b^t;g^t\right)}{1+\underset{s_V^G}{\rightarrow}\left(x^{t+1},y^{t+1},b^{t+1};g^{t+1}\right)/1+\underset{s_V^{t+1}}{\rightarrow}\left(x^{t+1},y^{t+1},b^{t+1};g^{t+1}\right)} \quad (2)$$

where the GML index can be decomposed into two components: technical efficiency (GEFFCH) and technical progress (GTECH), $x$, $y$, $b$, and $g$ denote input variables, consensual output variables, nonconsensual output variables, and directional variables, respectively, and $\underset{s_V^G}{\rightarrow}\left(x^t,y^t,b^t;g^t\right)$ denotes the full domain SBM directional distance function.

(2) Explanatory Variable

The pilot policy variable ($D$) for the national-level big data comprehensive pilot zone. This variable is a dummy variable for whether to establish a pilot zone and takes the value of 1 if province $i$ (including the province to which the pilot city of the pilot zone belongs) becomes a national-level big data comprehensive pilot zone in year $t$, and 0 otherwise.

(3) Control Variables

In order to accurately assess whether the development of the digital economy can promote the policy effect of the green transformation and upgrading of the manufacturing industry, this paper refers to the existing literature and controls the relevant factors that may have an impact on the policy assessment. Among them, the level of economic development ($lnpgdp$) is measured by per capita gross regional GDP; the level of opening to the outside world ($open$) is measured by the share of total imports and exports in regional GDP; the level of urbanization ($town$) is measured by the share of the urban population in year-end resident population; the level of R&D investment ($rdi$) is measured by the share of R&D investment in regional GDP measured; the level of human capital ($hc$) is measured using the average years of education by referring to Liu Da (2018) [38]. The calculation formula is as follows:

$hc$ = (number of people educated in elementary school × 6 + number of people educated in junior high school × 9 + number of people educated in high school or secondary school × 12 + number of people educated in college and above × 16)/total number of people aged 6 years and above.

(4) Mechanism Variables

Green technological innovation ($lngti$) is measured by the number of green invention patents granted; industrial structure ($is$) is measured by the proportion of tertiary industry value added in secondary industry value added; manufacturing agglomeration ($mag$) is measured by the location entropy index, and the calculation formula is as follows:

$$mag_{it} = \frac{MIV_{it}/MIV_t}{GDP_{it}/GDP_t} \quad (3)$$

where $mag_{it}$ denotes the manufacturing agglomeration of province $i$ in year $t$, $MIV_{it}$ denotes the total manufacturing output value of province $i$, $MIV_t$ denotes the national total manufacturing output value, $GDP_{it}$ denotes the regional GDP of province $i$, and $GDP_t$ denotes the national GDP.

(5) Threshold Variable

The environmental regulation intensity ($er$) is measured by referring to the method of Ren Xiaosong (2020) [39], using a comprehensive index of environmental regulation intensity calculated based on the emissions of three wastes per unit of output value, and the calculation formula is as follows:

Firstly, industrial wastewater emissions, industrial $SO_2$ emissions, and industrial smoke emissions per unit of output value are standardized and calculated as follows:

$$PE_{ij}^S = PE_{ij} - \min(PE_j)/\max(PE_j) - \min(PE_j) \tag{4}$$

where $PE_{ij}$ is the emission per unit output value of pollutant of category $j$ in province $i$, and $PE_{ij}^S$ is the standardized result of this indicator. $\max(PE_j)$ denotes the maximum value of emission per unit output value of pollutant of category $j$ in all provinces, $\min(PE_j)$ denotes the minimum value of emission per unit output value of pollutant of category $j$ in all provinces.

On this basis, the weights of various pollutants are calculated by the following formula:

$$W_j = PE_{ij}/\overline{PE_{ij}} \tag{5}$$

where $W_j$ denotes the weight of the $j_{th}$ pollutant, and $\overline{PE_{ij}}$ denotes the average level of emissions per unit of output value of the $j_{th}$ pollutant in province $i$.

Finally, the comprehensive index of environmental regulation intensity is calculated, and the formula is as follows:

$$er_{ij} = \frac{1}{3}\sum_{j=1}^{3} W_j PE_{ij}^S \tag{6}$$

The descriptive statistics of the above-related variables are shown in Table 1:

**Table 1.** Descriptive Statistics.

| Variables | Minimum | Maximum | Mean Value | Standard Deviation | Samples |
|---|---|---|---|---|---|
| $mgtfp$ | 0.170 | 4.979 | 1.442 | 0.857 | 510 |
| $D$ | 0 | 1 | 0.120 | 0.325 | 510 |
| $lnpgdp$ | 8.190 | 12.008 | 10.320 | 0.745 | 510 |
| $open$ | 0.013 | 1.711 | 0.320 | 0.376 | 510 |
| $town$ | 24.770 | 89.600 | 52.851 | 14.271 | 510 |
| $rdi$ | 1.783 | 5.781 | 4.353 | 0.633 | 510 |
| $hc$ | 6.176 | 13.829 | 9.443 | 1.293 | 510 |
| $lngti$ | 0 | 8.828 | 5.027 | 1.756 | 510 |
| $is$ | 0.156 | 4.979 | 1.401 | 0.582 | 510 |
| $mag$ | 2.787 | 4.472 | 3.860 | 0.282 | 510 |
| $er$ | 0.005 | 0.084 | 0.033 | 0.019 | 510 |

## 5. Empirical Analysis

Cross-sectional dependence is a key problem in testing the relationship between the selected variables in the panel data model. Ignoring it may lead to estimation bias and model failure [40]. Therefore, before the regression analysis, this research, referring to Yue Dou et al. (2021), first tested the cross-sectional dependence, mainly using the Breusch–Pagan LM and Pesaran CD tests [41]. The results are shown in Table 2; it can be seen that the statistics are significant at the 1% level, which proves that the cross-sectional dependence test passes.

**Table 2.** Cross-sectional Dependence Tests.

| Test | Statistics | Prob. |
|---|---|---|
| Breusch–Pagan LM test | 1620.79 *** | 0.0000 |
| Pesaran CD test | 12.35 *** | 0.0000 |

Note: *** indicate that they passed the significance test at the level of 1%.

### 5.1. Benchmark Regression

The results of the baseline regression of the impact of digital economy development on the green transformation and upgrading of the manufacturing industry are shown in Table 3,

where model (1) shows the results controlling for year-fixed effects and province-fixed effects only, while models (2)–(6) show the results controlling for year- and province-fixed effects with the inclusion of the control variables selected in this research one by one.

**Table 3.** Benchmark Regression.

| Variables | mgtfp | | | | | |
|---|---|---|---|---|---|---|
| | **(1)** | **(2)** | **(3)** | **(4)** | **(5)** | **(6)** |
| $D$ | 0.404 *** | 0.462 *** | 0.437 *** | 0.432 *** | 0.399 *** | 0.364 *** |
| | (0.105) | (0.095) | (0.087) | (0.090) | (0.086) | (0.085) |
| $lnpgdp$ | | 0.376 *** | 0.433 *** | 0.433 *** | 0.423 *** | 0.396 *** |
| | | (0.058) | (0.076) | (0.077) | (0.073) | (0.070) |
| $open$ | | | −0.009 | −0.012 | 0.003 | 0.017 |
| | | | (0.008) | (0.008) | (0.011) | (0.013) |
| $town$ | | | | 0.001 | 0.001 | 0.002 |
| | | | | (0.002) | (0.001) | (0.001) |
| $rdi$ | | | | | −0.496 | −0.448 |
| | | | | | (0.239) | (0.235) |
| $hc$ | | | | | | −0.541 |
| | | | | | | (0.288) |
| $constant$ | 1.394 *** | −2.165 *** | −2.205 *** | −2.156 *** | 2.230 | 1.445 |
| | (0.016) | (0.543) | (0.621) | (0.661) | (2.177) | (2.205) |
| Province effect | √ | √ | √ | √ | √ | √ |
| year effect | √ | √ | √ | √ | √ | √ |
| $N$ | 510 | 510 | 510 | 510 | 510 | 510 |
| $R^2$ | 0.825 | 0.848 | 0.849 | 0.850 | 0.857 | 0.859 |

Note: *** indicate that they passed the significance test at the level of 1%.

According to Table 3, we can see that the coefficients of the policy dummy variables are always significantly positive at the 1% level, regardless of whether the control variables are added or not, which indicates that the development of the digital economy can significantly promote the green transformation and upgrading of the manufacturing industry. Specifically, compared with nonpilot provinces, the establishment of the big data comprehensive pilot zone has led to an average increase of 36.4% in the green total factor productivity of the manufacturing industry in the pilot provinces, so Hypothesis 1 is valid. According to the control variables selected in this paper, the coefficient of per capita GDP is significantly positive, which indicates that the green total factor productivity of the manufacturing industry can be significantly improved with the improvement of economic development level, that is, economic development is conducive to promoting the green transformation and upgrading of the manufacturing industry. In addition, the coefficients of other control variables are not significant; that is, the level of opening to the outside world, the level of urbanization, the level of R&D investment, and the level of human capital have no significant impact on the green transformation and upgrading of the manufacturing industry.

### 5.2. Robustness Tests

(1) Parallel Trend Test

An important prerequisite for the application of the DID model is to satisfy the parallel trend hypothesis; that is, the time change trend of the treatment group and the control group should be consistent without policy influence. In this regard, this research refers to the method of Wang Banban (2020) [42] and constructs the following model using the event analysis method:

$$mgtfp_{it} = \beta_0 + \sum_{k \geq -8,\, k \neq 1}^{6} \beta_k D_{it}^k + \lambda X_{it} + \mu_i + \delta_t + \varepsilon_{it} \tag{7}$$

where $i$ denotes the province, $t$ denotes the year, and $D_{it}^k$ denotes the dummy variable for the period when the comprehensive pilot zone for big data was established. Suppose the policy time node of the test area is denoted by $e_i$, and if $t - e_i = k$, then $D_{it}^k = 1$, otherwise

$D_{it}^k = 0$. This research investigates the six periods after the establishment of the pilot zone, that is, $k \leq 6$, and the eight periods before the establishment of the pilot zone, namely $k \geq -8$. Meanwhile, this research takes the year before the establishment of the pilot zone as the baseline year, that is, $k \neq 1$. The coefficient $\beta_k$ indicates the impact on the green transformation and upgrading of the manufacturing industry before and after the establishment of the pilot zones, and if the coefficient is not significantly different from 0 when $k < 0$, it means that the parallel trend test is passed. The meanings of other variables are the same as in Equation (1). The results of the parallel trend test are shown in Figure 2, where the horizontal axis indicates the number of years before and after the establishment of the big data comprehensive pilot zone, and the vertical axis indicates the estimated coefficient of the policy treatment effect. It is not difficult to find that the estimated values of $\beta_k$. are not significantly different from 0 before the establishment of the pilot zone, indicating that the parallel trend hypothesis is satisfied; the treatment effect of the policy starts to be significant after the establishment of the pilot zone and shows an increasing trend year by year, indicating that the development of the digital economy can significantly promote the green transformation and upgrading of the manufacturing industry, which fully indicates that the regression results are robust.

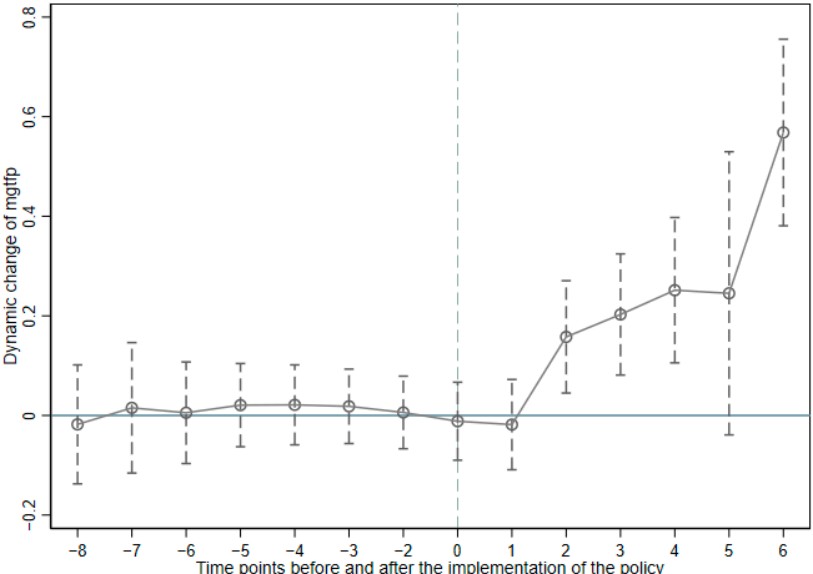

**Figure 2.** Parallel Trend Test.

(2) PSM-DID test

Randomness in the selection of treatment and control groups is another important prerequisite for the application of the double difference model, and this paper draws on the practice of Shi Daqian et al. (2018) to use the control variables as covariates and match the treatment and control groups using the nearest neighbor matching method to alleviate the selectivity bias problem [31]. The specific idea is as follows: the policy dummy variable is Logit regression to the control variable to calculate the tendency score. The calculation formula is as follows:

$$p_i(x_i) = P(D_i = 1 | x = x_i) = g(\varphi(x_i)) \tag{8}$$

where $p_i(x_i)$ is the propensity score, $D_i$ is the policy dummy variable, and the values of the control and experimental groups are 0 and 1, respectively, $\varphi(x_i)$ is the linear function of the control variables, and $g$ is the distribution function of Logit. Provinces with similar propensity score values were used as the control group to conduct a common trend test with the treatment group as a way to verify whether they satisfy the common support hypothesis, and the results are shown in Figure 3.

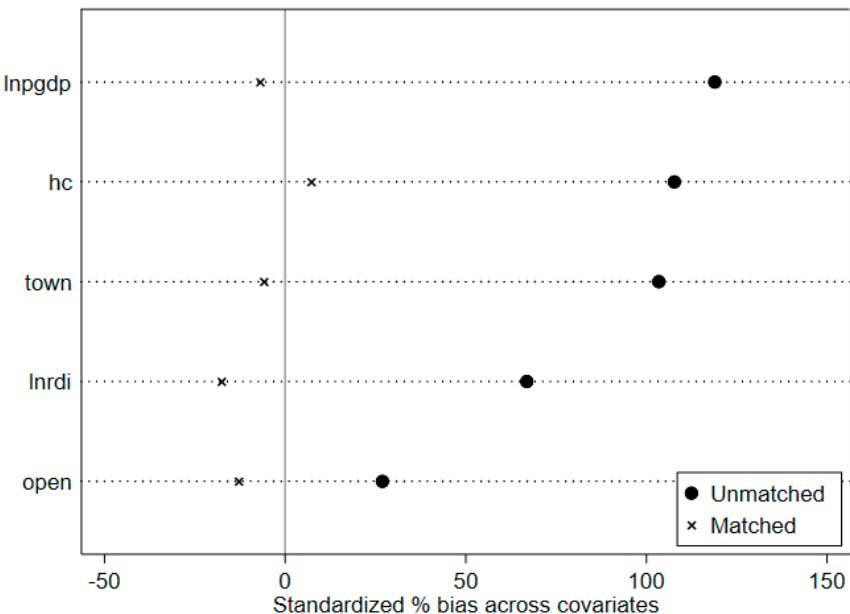

**Figure 3.** Common Trend Test.

It can be seen that there was no significant difference between the treatment and control groups before and after matching, and the standardized deviation after matching was significantly smaller, and the t-value was not significantly different from 0, which indicates that PSM is valid and DID analysis can be conducted on this basis. The results are shown in models (1) and (2) in Table 4, where the coefficient estimates of the policy dummy variables are significantly positive, which is basically consistent with the results of the baseline regression.

**Table 4.** Robustness Tests.

| Variables | PSM-DID Test | | Changing the Model Setting | | Substitution of Explanatory Variable | |
|---|---|---|---|---|---|---|
| | **(1)** *mgtfp* | **(2)** *mgtfp* | **(3)** *mgtfp* | **(4)** *mgtfp* | **(5)** *lnmai* | **(6)** *lnmai* |
| *D* | 0.281 *** | 0.303 *** | 0.413 *** | 0.377 *** | 0.213 *** | 0.206 *** |
| | (0.098) | (0.096) | (0.107) | (0.087) | (0.068) | (0.061) |
| *Control variables* | × | √ | × | √ | × | √ |
| *Province effect* | √ | √ | √ | √ | √ | √ |
| *year effect* | √ | √ | √ | √ | √ | √ |
| *constant* | 1.350 *** | −2.373 | 1.394 *** | 1.353 | 1.376 *** | 2.743 |
| | (0.015) | (2.532) | (0.016) | (2.216) | (0.014) | (2.761) |
| *N* | 420 | 420 | 510 | 510 | 510 | 510 |
| $R^2$ | 0.880 | 0.896 | 0.825 | 0.860 | 0.801 | 0.808 |

Note: *** indicate that they passed the significance test at the level of 1%.

(3) Placebo Test

Due to the inconsistent timing of the establishment of the big data comprehensive test area, this research refers to the method of Li Pei et al. (2016) for the placebo test [43]. The implementation steps are as follows: first, 10 provinces are randomly selected from 30 provinces as the pseudo-treatment group and other provinces as the control group; then, one year is randomly selected from the pseudo-treatment group as the pseudo-implementation year of the policy; finally, the pseudo-policy dummy variable is generated, and the regression is repeated 500 times as a way to test whether the establishment of the pilot zone has a significant effect on the randomly selected treatment group. The distribution diagram of processing effect estimates drawn is shown in Figure 4, from which we can see that the treatment effect estimates of the random group are concentrated around 0, with a large difference from the true coefficient estimates, and most of the *p*-values are

greater than 0.1, indicating that the establishment of the big data comprehensive pilot zone has no significant effect on the randomly selected pseudo-treatment group, which further illustrates the robustness of the regression results.

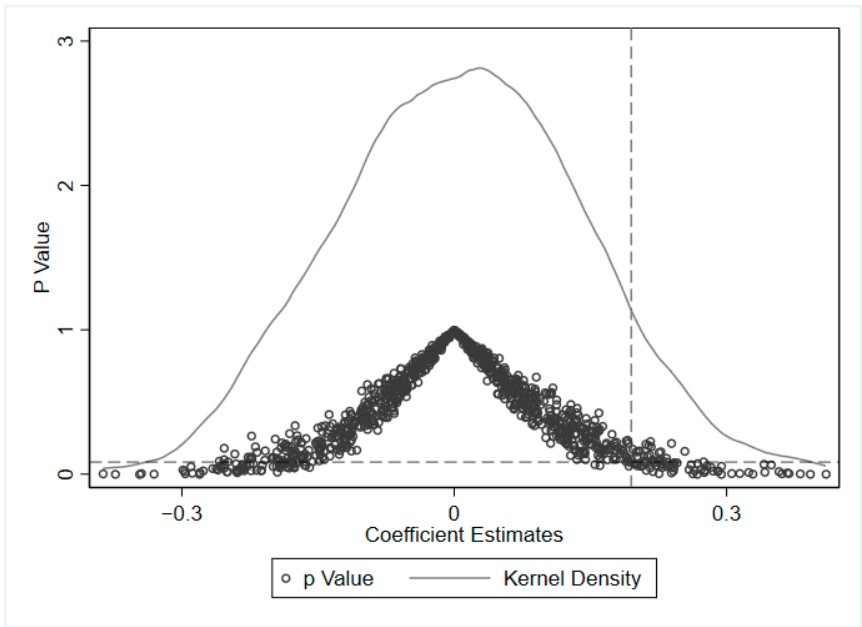

**Figure 4.** Placebo Test.

(4) Changing the Model Setting

Since the majority of big data comprehensive pilot zones were established in 2016, this paper refers to the method of Qiu Zixun (2021) [1], which sets 2016 as the policy starting year of big data comprehensive pilot zones, takes the year dummy variable of 2016 and subsequent years as 1 and the previous ones as 0; it takes the province dummy variable of pilot provinces in pilot zones as 1 and the nonpilot provinces as 0, and then the new policy dummy variables are generated by multiplying the two together and estimated using the traditional double difference model. The results are shown in models (3) and (4) in Table 4, and the coefficient of the interaction term is significantly positive, which is basically consistent with the baseline regression results.

(5) Substitution of Explanatory Variables

This paper refers to the practice of Zhang Feng et al. (2019) and introduces the directional distance function of slack variables to measure the green transition efficiency index (*lnmai*) of the manufacturing industry according to the industrial green growth analysis method and use it as a substitute variable for the explanatory variables for regression analysis [30]. The results are shown in models (5) and (6) in Table 4, where the coefficients of the policy dummy variables remain significantly positive and do not differ significantly from the baseline regression results.

*5.3. Heterogeneity Tests*

In order to test whether the establishment of the big data comprehensive pilot zone has different impacts on the green transformation and upgrading of manufacturing industries in different regions, this paper will further analyze the heterogeneity of the policy effects from two perspectives: the level of economic development and the level of network development.

(1) Level of Economic Development

According to the above benchmark regression results, economic development not only promotes the green transformation and upgrading of the manufacturing industry but also enhances the processing effect of the establishment of the big data comprehensive pilot zone on the green transformation and upgrading of the manufacturing industry.

Then, is there a difference in the treatment effect of the policy for regions with different economic development levels? Based on this, this paper refers to the practice of Qiu Zixun and Zhou Yahong (2021) [1] and divides the sample into two categories, developed and less developed regions, based on the criterion of whether the per capita GDP of each province in the previous period at the policy point in time is greater than the per capita GDP of the whole country in the current period, and conducts regression analysis separately. The results are shown in models (1) and (2) in Table 5. The results show that the promotion effect of the establishment of the pilot zone on the green transformation and upgrading of manufacturing industries in developed regions is not significant, while the promotion effect on less developed regions is very significant. The possible reason is that compared with developed regions, less developed regions pay more attention to the development opportunities of information and technology and realize the leap of their own economic development level through the integration and development of new industries and traditional advantageous industries.

**Table 5.** Heterogeneity Tests.

| Variables | *mgtfp* | | | |
|---|---|---|---|---|
| | **(1) Developed Regions** | **(2) Less Developed Regions** | **(3) Regions of High Network Development Level** | **(4) Regions of Low Network Development Level** |
| $D$ | 0.130 | 0.302 ** | 0.206 ** | 0.201 |
| | (0.123) | (0.132) | (0.010) | (0.155) |
| *Control variables* | √ | √ | √ | √ |
| *Province effect* | √ | √ | √ | √ |
| *year effect* | √ | √ | √ | √ |
| *constant* | 2.391 | −1.133 | 1.939 | −3.987 * |
| | (2.518) | (2.870) | (2.303) | (2.271) |
| $N$ | 289 | 221 | 340 | 170 |
| $R^2$ | 0.859 | 0.895 | 0.861 | 0.916 |

Note: ** and * indicate that they passed the significance test at the level of 5%, and 10%.

(2) Network Development Level

The development of the digital economy mainly relies on Internet technology, and the establishment of a big data comprehensive pilot zone requires the construction of perfect network infrastructure. Therefore, the promotion effect of digital economy development on the green transformation and upgrading of the manufacturing industry should be different for regions with different levels of network development. Based on this, this paper refers to the method of Guo Bingnan (2022) [4] and divides the sample into two categories of areas with high network development levels and areas with low network development levels according to the median number of international Internet users in each province in the period before the policy time point, and conducts regression analysis, respectively. The results are shown in models (3) and (4) in Table 5. The regression results show that the promotion effect of the establishment of the pilot zone on the green transformation and upgrading of the manufacturing industry in areas with high network development levels is more significant but has no significant promoting effect on the areas with low network development levels. The reason may be that compared with regions with lower network development levels, regions with higher network development levels can give full play to the advantages of the digital economy in promoting the green transformation and upgrading of the manufacturing industry.

*5.4. Testing the Influence Mechanism*

(1) Mediation Effect Test

On the basis of the previous verification that the development of the digital economy can promote the green transformation and upgrading of the manufacturing industry, further

verify the influence mechanism of the development of the digital economy to promote the green transformation and upgrading of the manufacturing industry through promoting green technological innovation and industrial structure upgrading from two perspectives of technological innovation effect and industrial structure effect, and construct the mediating effect model as follows:

$$media_{it} = \gamma_0 + \gamma_1 D_{it} + \gamma_2 X_{it} + \mu_i + \delta_t + \varepsilon_{it} \tag{9}$$

$$mgtfp_{it} = \delta_0 + \delta_1 D_{it} + \delta_2 media_{it} + \delta_3 X_{it} + \mu_i + \delta_t + \varepsilon_{it} \tag{10}$$

Among them, $media_{it}$ denotes the mediating variables, including green technology innovation ($lngti_{it}$) and industrial structure ($is_{it}$), which mainly focus on the significance of $\gamma_1$, $\delta_1$ and $\delta_2$, and other variables have the same meaning as equation (1). The results of the mediation effect test are shown in Table 6, where models (1) and (2) and models (3) and (4) test the mediation effects of green technological innovation and industrial structure, respectively. From them, we can see that some of the mediating effects are significant for both the technological innovation effect and the industrial structure effect. This indicates that the digital economy can realize the green transformation and upgrade of the manufacturing industry by promoting green technological innovation and industrial structure upgrade; therefore, Hypotheses 2a and 2b of the research are confirmed.

**Table 6.** Test of Influence Mechanism.

| Variables | Technological Innovation Effect | | Industrial Structure Effect | | Industrial Agglomeration Effect |
|---|---|---|---|---|---|
| | **(1)** *lngti* | **(2)** *mgtfp* | **(3)** *is* | **(4)** *mgtfp* | **(5)** *mgtfp* |
| D | 0.126 ** | 0.252 *** | 0.030 ** | 0.171 *** | 0.220 *** |
| | (0.0547) | (0.080) | (0.012) | (0.056) | (0.059) |
| lngti | | 0.554 *** | | | |
| | | (0.090) | | | |
| is | | | | 0.789 *** | |
| | | | | (0.224) | |
| mag | | | | | 0.152 ** |
| | | | | | (0.065) |
| $D \times mag$ | | | | | −0.059 ** |
| | | | | | (0.026) |
| Control variables | √ | √ | √ | √ | √ |
| Province effect | √ | √ | √ | √ | √ |
| year effect | √ | √ | √ | √ | √ |
| constant | 3.564 *** | −0.322 | −3.755 *** | −0.034 | −3.508 ** |
| | (0.649) | (1.107) | (0.322) | (1.546) | (1.587) |
| N | 510 | 510 | 510 | 510 | 510 |
| $R^2$ | 0.964 | 0.886 | 0.999 | 0.881 | 0.878 |

Note: ***, ** indicate that they passed the significance test at the level of 1%, 5%.

(2) Moderating Effect Test

According to the above benchmark regression results, economic development not only promotes the green transformation and upgrading of the manufacturing industry, but furthermore, this paper verifies the role of manufacturing agglomeration in the influence of digital economy development on the green transformation and upgrading of the manufacturing industry from the perspective of industrial agglomeration effect. Referring to the practice of Shi Dan (2020) [28], the moderating variable of manufacturing agglomeration as influencing the green transformation and upgrading of the digital economy is used to construct the moderating effect model as follows:

$$mgtfp_{it} = \alpha_0 + \alpha_1 D_{it} + \alpha_2 mag_{it} + \alpha_3 D_{it} mag_{it} + \alpha_4 X_{it} + \mu_i + \delta_t + \varepsilon_{it} \tag{11}$$

Among them, $mag_{it}$ denotes the moderating variable manufacturing agglomeration, $D_{it}mag_{it}$ is the cross-product term of policy dummy variable and moderating variable, we focus on the significance of coefficient $\alpha_3$, other variables mean the same as Equation (1). The results of the moderating effect test are shown in model (5) in Table 6, in which the coefficient of manufacturing agglomeration is significantly positive, indicating that manufacturing agglomeration can promote the green transformation and upgrading of the manufacturing industry; the coefficient of the interaction term between manufacturing agglomeration and the policy dummy variable is significantly negative, indicating that manufacturing agglomeration plays a negative moderating role in the influence of digital economy development on the green transformation and upgrading of the manufacturing industry, thus verifying the research Hypothesis 3.

*5.5. Threshold Effect Test*

In order to verify the role of environmental regulation in the impact of digital economy development on the green transformation and upgrading of the manufacturing industry, this paper refers to the practice of Yang Dan et al. (2020) and Yu Donghua et al. (2017), and takes the intensity of environmental regulation as the threshold variable to test the difference in the impact of the digital economy on the green transformation and upgrading of the manufacturing industry in different sections [32,44]. In this paper, the Bootstrap method was used to sample 300 times to estimate threshold values, and the results are shown in Table 7. The results show that neither the three thresholds nor the double thresholds are significant, while the single threshold value of 0.0074 is significant at the level of 10%, indicating that there is a single threshold effect on the intensity of environmental regulation, so Hypothesis 4 is valid.

**Table 7.** Significance Test of the Threshold Effect.

| *er* | Threshold | F Value | *p* Value | Bootstrap | 1% Threshold Value | 5% Threshold Value | 10% Threshold Value |
|---|---|---|---|---|---|---|---|
| Single | 0.0074 * | 24.72 | 0.0967 | 300 | 38.5235 | 29.5857 | 24.4176 |
| Double | 0.0078 | 5.24 | 0.7933 | 300 | 38.8048 | 30.5588 | 22.3206 |
| Triple | 0.0560 | 4.78 | 0.7567 | 300 | 31.6751 | 23.1950 | 18.4681 |

Note: * indicate that they passed the significance test at the level of 10%.

As to the analysis, this research refers to the method of Wang Ying (2021) [45] to construct a panel threshold model as follows:

$$mgtfp_{it} = \beta_0 + \beta_1 D_{it}(er_{it} \le \pi) + \beta_2 D_{it}(er_{it} > \pi) + \beta_3 X_{it} + \mu_i + \delta_t + \varepsilon_{it} \tag{12}$$

where $er_{it}$ denotes the intensity of environmental regulation as a threshold variable, $\pi$ denotes the single threshold value to be estimated, and other variables have the same meaning as in equation (1). According to the results of the threshold effect test and the existence of the single threshold value, the sample is divided into two, and the threshold effect regression is performed separately. In Table 8, the results show that when the environmental regulation is less than 0.0074, the coefficients of both the policy dummy variable and the intensity of environmental regulation are insignificant; when the intensity of environmental regulation is greater than 0.0074, the estimated coefficients of both the policy dummy variable and the intensity of environmental regulation are significantly positive. This indicates that when the increase in environmental regulation intensity breaks the threshold value, the promotion effect of digital economy development on the green transformation and upgrading of the manufacturing industry is significantly increased.

**Table 8.** Threshold Effect Regression.

| Variables | Environmental Regulation Effect | |
|---|---|---|
| | **(1)** *mgtfp* | **(2)** *mgtfp* |
| *D* | 0.050 | 0.169 *** |
| | (0.344) | (0.058) |
| *er* ≤ 0.0074 | −0.180 | |
| | (0.133) | |
| *er* > 0.0074 | | 0.784 ** |
| | | (0.097) |
| *Control variables* | √ | √ |
| *Province effect* | √ | √ |
| *year effect* | √ | √ |
| *constant* | 6.979 | −3.300 ** |
| | (4.876) | (1.579) |
| *N* | 49 | 461 |
| $R^2$ | 0.904 | 0.887 |

Note: ***, ** indicate that they passed the significance test at the level of 1%, 5%.

## 6. Conclusions and Recommendations

Based on the panel data of the manufacturing industry of 30 provincial administrative units from 2005 to 2021, this paper empirically analyzes the processing effect of digital economy development on the green transformation and upgrading of the manufacturing industry by using the quasi-natural experiment of the establishment that follows the national big data comprehensive pilot zone. The conclusions are as follows: First, the development of the digital economy can significantly promote the green transformation and upgrading of the manufacturing industry. The establishment of the big data pilot zone greatly promotes the development of the local digital economy, thus speeding up the pace of the green transformation and upgrading of the manufacturing industry. Second, the development of the digital economy plays a significant role in the green transformation and upgrading of the manufacturing industry in economically underdeveloped regions and network-developed regions; the enabling role of the digital economy in the green transformation and upgrading of the manufacturing industry can be more effectively played. Third, the digital economy can stimulate the green technology innovation of manufacturing enterprises and promote the optimization of the manufacturing industry structure so as to realize the green transformation and upgrading of the manufacturing industry. Fourth, the agglomeration of the manufacturing industry will have a regulating effect on the impact of the digital economy on the green transformation and upgrading of the manufacturing industry. Fifth, the higher intensity of environmental regulation can significantly enhance the promotion effect of the digital economy on the green transformation and upgrading of the manufacturing industry so as to utilize the policy effect of big data pilot zone on the green transformation and upgrading of the manufacturing industry better.

Based on the above conclusions, suggestions are put forward as follows: First, strengthen the promotion and construction of the big data comprehensive pilot zone to further promote the construction and development of the digital economy. On the one hand, implementing the development mode of combining the emerging big data industry with the traditional advantageous industry can help the manufacturing industry to realize the transformation of "getting rid of the virtual and real"; on the other hand, traditional industries should be encouraged to make full use of digital economy to realize transformation and upgrading, and adopt the green development mode with both economic and environmental benefits, which is conducive to the sustainable development of traditional industries. Second, the regions with relatively backward economic development should pay more attention to the construction of network infrastructure and give full play to the advantages of the digital economy, enabling the green transformation and upgrading of the manufacturing industry by relying on the Internet. We should firmly grasp the opportunity of setting up the big data pilot zone and take the development of the digital economy as

the general focus to promote the green transformation and upgrading of the manufacturing industry. Third, introduce corresponding policies to encourage enterprises to carry out green technological innovation and guide manufacturing enterprises to take the path of green transformation and upgrading. Fourth, rational planning of the manufacturing agglomeration area gives full play to the role of manufacturing agglomeration and avoids the negative impact of excessive agglomeration on the green transformation and upgrading of the manufacturing industry. Fifth, strengthen environmental regulations, rationally plan and scientifically implement regulatory measures, ensure the depth and breadth of environmental regulations, and help the development of the digital economy to promote the green transformation and upgrading of manufacturing.

**Author Contributions:** Conceptualization, Q.G.; methodology, Q.G.; software, Q.G. and X.W.; validation, Q.G. and X.W.; formal analysis, Q.G. and X.W.; investigation, X.W.; resources, X.W.; data curation, X.W.; writing—original draft preparation, Q.G. and X.W.; writing—review and editing, X.T.; visualization, X.T.; supervision, Q.G.; project administration, Q.G.; funding acquisition, Q.G. All authors have read and agreed to the published version of the manuscript.

**Funding:** This research was funded by National Social Science Fund grant number 18BGL173, Shaanxi Provincial Social Science Fund grant number 2020R038, and Xi'an Shiyou University Graduate Innovation and Practical Capability Project Fund grant number YCS22214288.

**Institutional Review Board Statement:** Not applicable.

**Informed Consent Statement:** Not applicable.

**Data Availability Statement:** Publicly available datasets were analyzed in this study. This data can be found here: China Statistical Yearbook (2006–2020), Chinese Provinces Statistical Yearbooks (2006–2020), China Environmental Statistical Yearbook, China Energy Statistical Yearbook, EPS Database, and WIND Database.

**Conflicts of Interest:** The authors declare no conflict of interest.

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
