# Peer review of "How Can the Development of Digital Economy Empower Green Transformation and Upgrading of the Manufacturing Industry?—A Quasi-Natural Experiment Based on the National Big Data Comprehensive Pilot Zone in China"

_sustainability, doi:10.3390/su15118577_

Round 1

Reviewer 1 Report

The topic of the article is highly topical and interesting. However, the analyzed data series is relatively outdated and does not reflect modern trends and the advent of new technologies, nor does it reflect and compare similar trends, for example, the European Green Deal.

Even though the data series ends in 2019, the authors appropriately compare the findings with other, later research in the literature review (e.g. [23]). In the analysis of the existing literature, the authors insufficiently identified the factors and evaluation criteria, the authors mainly mention general factors: "factors such as innovation environment and entrepreneurship", "factors such as capital, talents and technology" (used in hypothesis 1), in hypothesis 2b the authors mention "manufacturing enterprises and their related production factors". The authors also state that data sources are one of the decisive production factors.

In their own work, however, they do not work with generally identified factors, they work mainly with Green total factor green productivity, which is not sufficiently justified.

Other factors are defined in Chapter 4.3.

Hypotheses are formulated and analyzed appropriately, but I lack conclusions related to hypothesis 4.

Recommendation:

Complete the conclusions related to hypothesis 4 specifically.

Specify the Green total factor green productivity option in more detail.

Include in the discussion a comparison with other activities, for example with the European Green Deal

Literature [25] is not cited in the text.

Reviewer 2 Report

This study is very interesting and related to what is required now a days. But while comparing the methods and results, reader tends to get lost in the process as in the methods only first step of estimation is explained while in results further investigation are conducted. 

Title is too long, it loses focus on what the study is addressing. Author need to improve the title while considering the objectives set by the study, any other information can be explained at its appropriate place. 

the data seems to be quite old till 2019. it is 2023 now. Author need to update the data or atleast compare the forecasts of the study with the new data to see how it is performing. 

At the end of section 2, some of the citations are not properly cited in the said format. 

Recent studies justifying the need for this specific estimation method are to be extended.

descriptive stats only shown mean and standard deviation, what about skewness, kurtosis and normality. would your estimation model be effected if it is found that the data is not normal. 

Author need to check for post regression diagnostics for the fixed effect models. These models are invalid in the presence of cross-sectional dependence. 

Author can add the moderator effect graphs to explain the changes visually. 

Similarly there is a need to add visualization of the data to check how the data is behaving so that it can be related to the outcomes. 

Authors used complex estimation methods, they are required to add simplified explanation of why these methods are necessary as the methodology section does not narrate what process that this study has taken.

in the implications author also need to provide methodological implications in terms of what is the role of this estimation method in achieving these results, what are the assumptions that are required to ensure that these results are reliable. and what future studies can ensure to make sure that they can achieve what this study has not. 

proofread can improve the quality 

Round 2

Reviewer 2 Report

In several places both APA and numbered citation are coming together, as per rule only numbered reference must be present. 
